# Response to anxiety treatment before, during, and after the COVID-19 pandemic

**David H. Rosmarin**[1,2]*, **Steven Pirutinsky**[2,3]

**1** McLean Hospital/Harvard Medical School, Belmont, MA, United States of America, **2** Center for Anxiety, New York, NY, United States of America, **3** Touro College, Graduate School of Social Work, New York, NY, United States of America

* drosmarin@mclean.harvard.edu

## Abstract

### Background

The COVID-19 pandemic yielded a substantial increase in worldwide prevalence and severity of anxiety, but less is known about effects on anxiety treatment.

### Objective

We evaluated effects of the COVID-19 pandemic on responses to Cognitive Behavioral Therapy for anxiety, in a clinically heterogeneous sample of patients.

### Methods

A sample of 764 outpatients were separated into four groups: (1) *Pre-pandemic* (start date on or prior to 12/31/2019), (2) *Pandemic-Onset* (start date from 01/01/2020 to 03/31/2020), (3) *During-Pandemic* (start date from 04/01/2020 through 12/31/2020), and (4) *Post-Pandemic* (start date on or after 01/01/2021). We subsequently compared treatment trajectories and effects within and between these groups over 5621 total time points (mean of 7.38 measurements per patient).

### Results

Overall, patients presented with moderate levels of anxiety ($M = 13.25$, 95%CI: 12.87, 13.62), which rapidly decreased for 25 days ($M = 9.46$, 95%CI: 9.09, 9.83), and thereafter slowly declined into the mild symptom range over the remainder of the study period ($M = 7.36$, 95%CI: 6.81, 7.91), representing clinically as well as statistically significant change. A series of conditional multilevel regression models indicated that there were no substantive differences between groups, and no increase in anxiety during the acute pandemic phase.

### Conclusions

Our results suggest that responses to treatment for anxiety were equivalent before, during, and after the COVID-19 pandemic. Among patients who were in treatment prior to the pandemic, we failed to detect an increase in anxiety during the pandemic's acute phase (March 20th, 2020 through July 1st, 2020).

**Data Availability Statement:** Data are publicly available from OSF at https://osf.io/yusdk/.

**Funding:** The authors received no specific funding for this work.

**Competing interests:** The authors have declared that no competing interests exist.

## Introduction

It is well established that the COVID-19 pandemic led to adverse effects on mental health for the population as a whole [1–5], and vulnerable subgroups in particular [6, 7]. Anxiety demonstrably increased substantially from the pandemic's onset in early 2020 [8–10], through the first availability of vaccinations in early 2021 (dubbed by some as the "light at the end of the tunnel") [11]. To quantify these trends: One large meta-analytic study with over two million adults found that 35% had significant anxiety during the pandemic [2], and the World Health Organization estimated a 25% increase overall [12, 13]. These effects are not surprising given that intolerance of uncertainty–which was rampant during the pandemic given high levels of perceived threat–is a key factor in the development and severity of anxiety [14, 15].

Less is known about the effects of the pandemic on *treatment* for anxiety. On the one hand, there is reason to believe that COVID-19 was detrimental to anxiety treatment. During 2020, several helpful commentaries and case reports were rapidly published to provide clinicians with specific strategies to support patients with pre-existing anxiety by bolstering treatment delivery [16, 17]. In retrospect, these efforts were well-warranted since it is now known that a history of mental health treatment prior to the pandemic predicted greater likelihood of having symptoms meeting criteria for Generalized Anxiety Disorder during the pandemic [18]. On the other hand, one study found that individuals who received Cognitive Behavioral Therapy (CBT) for social anxiety disorder *prior* to the pandemic, benefited from enduring effects [19]. Similarly, in another study of patients with severe obsessive-compulsive disorder, the trajectory and outcomes of intensive CBT was similar among those receiving treatment prior to vs. during the pandemic [20]. These findings are encouraging, and may suggest that the acquisition of cognitive and behavioral skills is a harbinger of better mental health, even in the context of uniquely high worldwide stress. However, further research in more clinically diverse samples is needed to assess whether CBT for anxiety was equally effective for those who entered treatment during the pandemic, compared to before, or after.

We therefore evaluated responses to anxiety treatment before, during, and after the pandemic, in a clinically heterogeneous sample of patients presenting to a naturalistic outpatient setting. We separated patients into four groups, in accordance with a COVID-19 pandemic timeline proposed by the Yale School of Medicine [21]: (1) *Pre-pandemic*: Those who entered and completed treatment before the start of the pandemic (start date on or prior to 12/31/2019); (2) *Pandemic-Onset*: Those who were in treatment during the onset of the pandemic (start date from 01/01/2020 to 03/31/2020); (3) *During-Pandemic*: Those who commenced treatment after the onset of the pandemic (start date from 04/01/2020 through 12/31/2020); and (4) *Post-Pandemic*: Those who entered treatment once vaccines started to become available (start date after 01/01/2021). Subsequently, we assessed and compared treatment trajectories and effects within and between these groups. We also examined whether patients in treatment during the pandemic experienced any specific changes in anxiety during the initial acute phase of COVID-19 (March 20[th] 2020 through July 1[st] 2020). We hypothesized that patients presenting to treatment prior to (group 1) and after the pandemic (group 4) would benefit more from treatment than those who received treatment during the pandemic's onset or prior to the availability of vaccines (groups 2 and 3). We further hypothesized that anxiety would worsen during the initial acute phase of the pandemic.

## Materials and methods

### Procedures & participants

Data was collected from adult patients presenting to the offices of Center for Anxiety, a multi-site outpatient clinic in the northeastern United States between 10/1/2019 and 3/1/2021. The study was approved by the Touro University Institutional Review Board for the Protection of

Human Subjects, protocol # IRB1-2023-003. At treatment intake, patients provided written informed consent to have data from their clinical questionnaires and medical records used in research. Medical record data was assessed retrospectively and was fully de-identified prior to access by the study team. At intake and at each treatment session, patients were asked to complete self-report measures of anxiety using Psych-Surveys™ software. At intake, patients also received a general psychosocial interview, as well as the Miniature International Neuropsychiatric Interview [22]. Inclusion criteria for the current study included age 18 years or older, and completion of anxiety measure at intake plus at least three additional times within the first 100 days treatment. We included only measurements that took place within the first 100 days of treatment, since measurements post 100 days were highly variable and sparse; this resulted in the exclusion of only 0.3% of patients. Our final sample included 764 patients, with anxiety assessed at 5621 total time points, representing a mean average 7.38 anxiety measurements per patient. Group sizes were as follows: *Pre-pandemic* (*n* = 221), (2) *Pandemic-Onset* (*n* = 42), (3) *During-Pandemic* (*n* = 104), (4) *Post-Pandemic* (*n* = 384).

All patients were provided with Cognitive-Behavioral Therapy (CBT) and/or Dialectical Behavior Therapy (DBT) as per usual clinic procedures. While no standardized treatment protocols were used given the naturalistic setting, a chart review revealed that a variety of specific cognitive and dialectical behavior therapy techniques were utilized including psychoeducation, monitoring of symptoms/target behaviors (e.g., thought records, diary cards), exposure, response prevention, behavioral activation, identifying and restructuring cognitive distortions, as well as mindfulness and acceptance. Therapists included doctoral level trainees as well as master's level clinicians, all of whom received weekly supervision and additional consultation as needed throughout treatment by a licensed provider. This study was approved by the Touro University Institutional Review Board for the Protection of Human Subjects.

## Measures

Demographic information was collected from electronic health records, and obtained from patients using a combination of self-report items and a semi-structured interview at intake.

Diagnoses were assessed with Miniature International Neuropsychiatric Interview [22], and conferred by supervising licensed doctoral-level staff.

Levels of anxiety were assessed at intake and each subsequent session using the GAD-7, a seven-item self-report measure of generalized anxiety symptoms that is used to assess for anxiety in a variety of clinical settings [23]. The scale yields a single total score between 0 and 21 and can be interpreted using four validated levels of anxiety severity: "Minimal" (0–4) "Mild" (5–9); "Moderate" (10–14) and "Severe" (15–21) [23].

## Statistical analyses

Given unequal group sizes, we modeled changes in anxiety over the course of treatment using multilevel growth curve models [24], which are widely used in psychotherapy research since they are robust, allow for missing data, handle designs with varying measurement times, and control for unmeasured between-subject differences. Models were estimated with the *lme4 library* [25] using restricted maximum likelihood estimation, and coefficients tested with the *lmerTest library* [26] in the R programming language [27]. Non-linear terms were constructed using Orthogonal Polynomials estimated by the poly function in the stats package [27], plots were created using the *sjPlot library* [28]. Descriptive statistics and preliminary analyses were calculated in SPSS 23.

Power analyses for longitudinal multilevel regression models require complex simulations with extensive assumptions to provide accurate estimates of power [29]. Given the complex nature of our analyses, we were unable to develop reasonable assumptions. However, previous

simulation studies indicate that multilevel modeling is highly robust and yields unbiased estimates of fixed effects even in small samples (e.g., as many as 10 groups with as few as five units each [30], and that these models generally require few cases to have sufficient power (e.g., as many as 50 groups with as few as five observations each [31]). Following these heuristics, we estimated that the current sample was likely to capture even small effects.

# Results

## Preliminary analyses

Demographic and clinical characteristics of each group within the sample are presented in Table 1. Results indicated that groups did not differ significantly in terms of any demographic variables. Subsequent analyses therefore did not include demographic covariates. Similarly, or diagnoses and levels of anxiety at intake were also statistically equivalent between groups. Groups also had equivalent numbers of weekly sessions, suggesting that treatment was not more or less intensive for any particular group.

## Treatment effects

Examination of raw treatment trajectories (Fig 1) suggested that changes in anxiety over the course of treatment was best described by a cubic pattern, with an initial period of rapid

**Table 1. Demographic & clinical characteristics of the sample.**

| Variable | Pre-pandemic (on or before 12/31/2019) | Pandemic-Onset (01/01/2020-03/31/2020) | During-Pandemic (04/01/2020-12/31/2020) | Post-Pandemic (1/1/2021 and thereafter) | Test Statistics |
|---|---|---|---|---|---|
| n | 221 | 42 | 104 | 384 | |
| Demographic Characteristics | | | | | |
| Age M(SD) | 30.67 (10.68) | (26.42) 7.87 | 35.34 (15.47) | 31.92 (13.77) | $F_{(3,550)} = 3.79$ |
| Gender (female) | 63% | 71% | 56% | 61% | $\chi^2_{(6,598)} = 15.90$ |
| Marital Status | Single 64% | Single 76% | Single 58% | Single 67% | X2(15,600) = 13.01 |
| | Married 28% | Married 24% | Married 29% | Married 25% | |
| | Sep/Div 5% | Sep/Div 0% | Sep/Div 7% | Sep/Div 4% | |
| | Cohab 3% | Cohab 0% | Cohab 6% | Cohab 3% | |
| Household size M (SD) | 3.11 (1.55) | 2.88 (1.61) | 2.93 (1.58) | 2.95 (1.54) | $F_{(3,596)} = .62$ |
| College Graduate | 63% | 67% | 63% | 64% | $\chi^2_{(12,600)} = 14.12$ |
| Unemployed | 11% | 7% | 13% | 11% | $\chi^2_{(21,600)} = 26.29$ |
| Clinical Characteristic | | | | | |
| Anxiety | M = 13.52 | M = 14.38 | M = 13.33 | M = 13.10 | $F_{(3, 598)} = .75$ |
| M (SD) | SD = 5.18 | SD = 5.20 | SD = 5.79 | SD = 5.35 | |
| Diagnoses | Anxiety 66% | Anxiety 50% | Anxiety 62% | Anxiety 64% | $\chi^2_{(3, 554)} < 8.4$ |
| | OC 23% | OC 8% | OC 20% | OC 23% | for all analyses |
| | Mood 39% | Mood 50% | Mood 45% | Mood 41% | |
| | Other 12% | Other 16% | Other 18% | Other 30% | |
| Weekly Sessions | .86 (.52) | .92 (.39) | .92 (.53) | .84 (.44) | $F_{(3, 748)} = 1.11$ |

Notes: All tests were not significant (p-level adjusted for multiple comparisons); n differs slightly between analyses due to missing data for some patients; Unemployed excludes homemakers, students, and retirees; Anxiety refers to GAD-7 scores at intake; Diagnoses sum to more than 100% as some patients presented with multiple concerns.

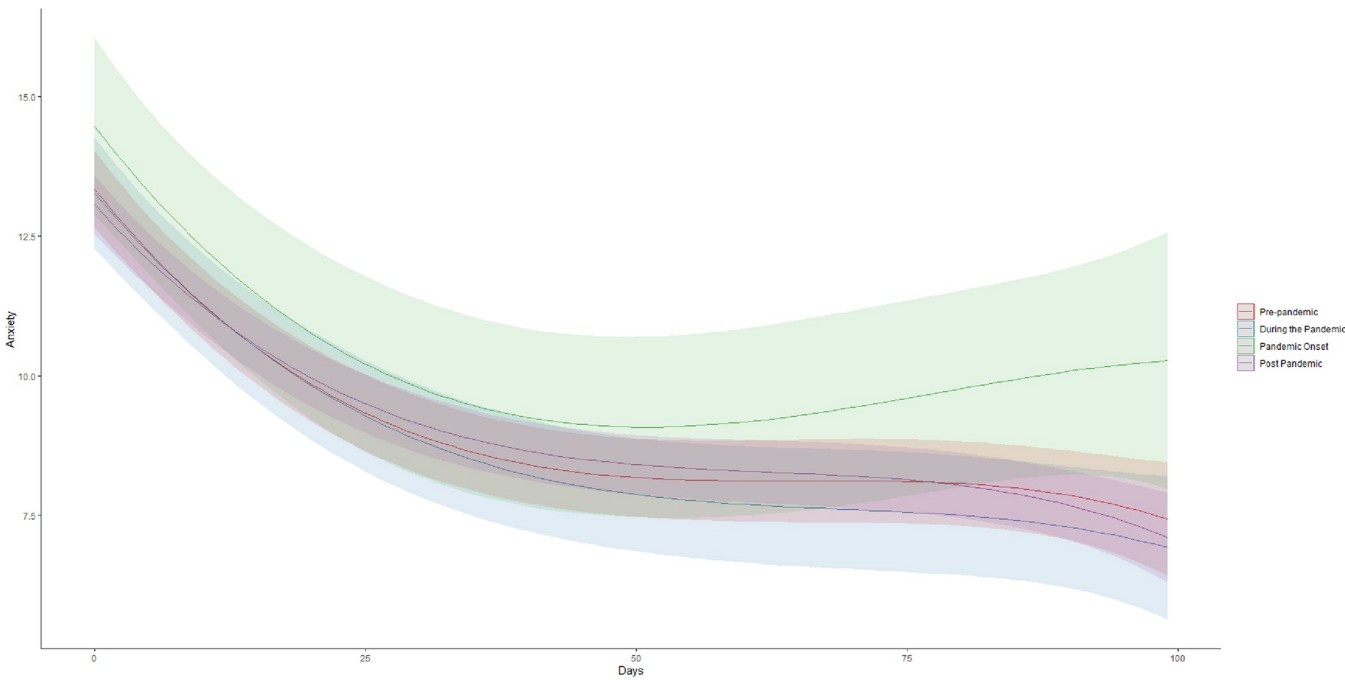

**Fig 1. Changes in anxiety before, during, and after the COVID-19 pandemic.**

decline lasting roughly 25 days followed by a longer period of slower improvement that slowly trailed off over roughly 75 days. Consistent with these descriptive data and previous research [32] results indicated that a model allowing for individual random variation in the linear and non-linear rates of change was the best fit for our data (Table 2). Specifically, cubic models fit significantly better than simpler linear models ($\Delta AIC$ = -10.99, $\Delta BIC$ = -5.36, $\chi^2(1)$ = 12.99, $p$ = .0003), quadratic models ($\Delta AIC$ = -10.99, $\Delta BIC$ = -5.36, $\chi^2(1)$ = 12.99, $p$ = .0003) and log-linear models ($\Delta AIC$ = -10.99, $\Delta BIC$ = -5.36, $\chi^2(1)$ = 12.99, $p$ = .0003). Coefficients for a baseline cubic treatment model are presented in Table 3. These indicate that on average, patients presented with moderate levels of anxiety ($M$ = 13.25, 95%CI: 12.87, 13.62), which rapidly decreased for 25 days ($M$ = 9.46, 95%CI: 9.09, 9.83), and thereafter slowly declined into the mild symptom range over the remainder of the study period ($M$ = 7.36, 95%CI: 6.81, 7.91).

**Table 2. Multilevel regression models of anxiety over the course of clinical treatment.**

|  | df | AIC | BIC | -2LL | $X^2$ | p |
|---|---|---|---|---|---|---|
| **Baseline Models** |  |  |  |  |  |  |
| M0: Intercept Only | 3 | 319009 | 31929 | -15951 |  |  |
| M1: Linear Time | 6 | 30424 | 30463 | -15206 | 1491.27 | < .0001 |
| M2: Quadric Time | 10 | 29739 | 29806 | -14860 | 692.31 | < .0001 |
| M3: Cubic Time | 15 | 29508 | 29608 | -14739 | 241.17 | < .0001 |
| **Conditional Models** |  |  |  |  |  |  |
| M4: Intake period (Intercept only) | 18 | 29511 | 29630 | -14737 | 3.46 | 0.33 |
| M5: Intake period (Slopes) | 27 | 29521 | 29700 | -14733 | 7.80 | 0.55 |
| M6: Acute Pandemic (Intercept) | 28 | 29522 | 29708 | -14733 | .62 | 0.43 |

Notes: All models were based on 764 patients and 5621 observations and including random intercepts and slopes for each patient

**Table 3. Unconditional multilevel regression models (treatment effects).**

| Fixed Effects | B | SE | t | p |
|---|---|---|---|---|
| Intercept | 9.65 | 0.17 | 57.76 | < .00001 |
| Linear Time | -118.85 | 5.30 | 22.42 | < .00001 |
| Quadratic Time | 61.36 | 4.28 | 14.32 | < .00001 |
| Cubic Time | -26.60 | 3.62 | 7.34 | < .00001 |
| Random Effects | SD | Linear | Quadratic | Cubic |
| Intercept | 4.43 | .03 | -.26 | .08 |
| Linear Time | 107.50 | | -.21 | -.30 |
| Quadratic Time | 84.83 | | | -.54 |
| Cubic Time | 2.31 | | | |

Notes: Model based on n = 764 patients and 5621 observations; Time represents the number of days since intake and was coded using Orthogonal Polynomials.

These results represent both clinically as well as statistically significant change in the sample as a whole.

## Pandemic effects

Building upon the above baseline model, we estimated a series of conditional multilevel regression models to assess if the course of anxiety differed between the above-mentioned four groups in our sample: (1) *Pre-pandemic*, (2) *Pandemic-Onset*, (3) *During-Pandemic*, (4) *Post-Pandemic*. Model comparisons are reported in Table 3 and indicate that there were no substantive differences between these groups: All entered treatment with roughly the same (moderate) levels of anxiety, all progressed through treatment in a similar cubic pattern, and all terminated with similar (mild) levels of anxiety. These results suggest that responses to psychotherapy for anxiety were equivalent before, during, and after the COVID-19 pandemic. Furthermore, among patients who were in treatment at the start of the pandemic (groups 2 and 3), an additional model assessing whether levels of anxiety increased during the initial acute phase of COVID-19 (March 20th 2020 through July 1st 2020) was similarly non-significant, suggesting that existing patients did not experience increased in anxiety over that time (Table 2, Model M6).

## Discussion

In this study, we examined effects and trajectories of anxiety treatment within a large and clinically diverse sample of patients presenting prior to, during, and after the COVID-19 pandemic. Contrary to our expectations, the course of anxiety and its treatment effects were equivalent among patients, irrespective of when they entered treatment. That is, irrespective of when patients commenced or terminated treatment, they had roughly the same levels of anxiety at the start of treatment, they then experienced a cubic pattern of anxiety change characterized by an initial period of rapid decline lasting roughly 25 days, followed by a longer period of slower improvement that slowly trailed off over roughly 75 days, and treatment resulted in similar levels of anxiety 100 days after patients' initial sessions. These results are consistent with a large body of literature highlighting the efficacy and effectiveness of CBT for anxiety-related concerns (e.g., [33–35]), which includes several studies demonstrating large, stable, and enduring effects [36–38]. Our findings support and extend this work by suggesting that treatments for anxiety are effective, even in the context of uniquely heightened periods of prolonged stress. While treatment was not standardized, the naturalistic setting of our study has

ecological validity and highlights the real-world value of CBT and DBT, even when delivered under unusual conditions.

Of potentially even greater significance, we found that pre-pandemic patients did *not* experience a discernable increase in anxiety during the initial acute phase of COVID-19. As noted above, this period of time was marked by significant mental distress [8–10] due to intense uncertainty, strain, and social isolation. The initiation of the COVID-19 pandemic was likely of particular concern for anxiety treatment-seekers, who experienced the additional stress of shifting to 100% telehealth in delivering their treatment, over the span of just a few weeks. Our findings optimistically suggests that the acquisition of psychosocial skills is a key predictor of mental health, wellbeing, and resilience–those who acquire such skills can benefit, even when facing to significant life stressors. Our findings also suggests that, ironically, those who experienced anxiety prior to the pandemic and took the opportunity to enter treatment, may have been better off than others who had never experienced significant anxiety before the pandemic. In this regard, previous experience of anxiety leading to treatment may reduce future susceptibility to symptoms in the context of increased stress.

Our study has several limitations that should be noted. First, our sample was demographically and clinically diverse, but highly educated overall and geographically specific to the northeastern United States. Treatment effects over the course of the pandemic might have been different within other regions, countries, or populations. Second, while multilevel modeling is robust to differences in group size, the *pandemic-onset* group was substantially smaller than the others. While this likely represents a smaller date range for group 3, it may also reflect that timing of psychosocial challenges and limited availability of in-person services can preclude entry into treatment. Our results should therefore not be construed to represent or reflect aggregate effects of stressors on anxiety overall, rather effects of the pandemic on the course and effects of symptoms among treatment-seekers. Finally, our study is limited to an analysis of anxiety, and treatment effects on other symptoms such as depression and substance abuse–both of which increased substantially during the pandemic [39, 40]–might have varied over the course of 2020.

In sum, our results indicate that response to anxiety treatment was strikingly similar for patients presenting before, during, and after the COVID-19 pandemic, suggesting that acquisition of skills to cope with anxiety is protective even in the context of a global crisis.

## Author Contributions

**Conceptualization:** David H. Rosmarin, Steven Pirutinsky.

**Data curation:** Steven Pirutinsky.

**Formal analysis:** Steven Pirutinsky.

**Funding acquisition:** David H. Rosmarin.

**Investigation:** David H. Rosmarin, Steven Pirutinsky.

**Methodology:** David H. Rosmarin, Steven Pirutinsky.

**Project administration:** David H. Rosmarin.

**Resources:** David H. Rosmarin.

**Supervision:** David H. Rosmarin.

**Writing – original draft:** David H. Rosmarin, Steven Pirutinsky.

**Writing – review & editing:** David H. Rosmarin, Steven Pirutinsky.

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
