## [Decision Letter · Decision Letter 0]

4 Sep 2023

PONE-D-23-19350Response to Anxiety Treatment Before, During and After the COVID-19 PandemicPLOS ONE

Dear Dr. Rosmarin,

Thank you for submitting your manuscript to PLOS ONE. After careful consideration, we feel that it has merit but does not fully meet PLOS ONE’s publication criteria as it currently stands. Therefore, we invite you to submit a revised version of the manuscript that addresses the points raised during the review process.

We look forward to receiving your revised manuscript.

Kind regards,

Mohammad Farris Iman Leong Bin Abdullah, Dr Psych

Academic Editor

PLOS ONE

Reviewers' comments:

Reviewer's Responses to Questions

**Comments to the Author**

1. Is the manuscript technically sound, and do the data support the conclusions?

Reviewer #1: Yes

Reviewer #2: Yes

2. Has the statistical analysis been performed appropriately and rigorously? 

Reviewer #1: I Don't Know

Reviewer #2: I Don't Know

3. Have the authors made all data underlying the findings in their manuscript fully available?

Reviewer #1: Yes

Reviewer #2: Yes

4. Is the manuscript presented in an intelligible fashion and written in standard English?

Reviewer #1: Yes

Reviewer #2: Yes

5. Review Comments to the Author

Reviewer #1: Please adding more relevant key words

Please consistently use same format in describing the dates such as July 1st 2020 throughout the article.

The result and discussion only highlighted about the efficacy of CBT for anxiety but in the methodology indicated that the respondents also received DBT.

Please mention how many sessions your respondents received and is there any standardized protocol to for your intervention. Please discuss the medium that you used in conducting your intervention.

Please explain how you compare the unequal group size and controlled the possible bias.

Reviewer #2: The role of CBT in anxiety treatment is an important area of research in mental health studies. The present article will add to the research in this area.

There are a few clarifications needed from the authors:

1. Was there uniformity of treatment in all groups? Did the "during pandemic" and "post pandemic " group receive any additional counselling, specific to the pandemic? Was their counselling more frequent?

2. The data in this study is of anxiety patients who adhered to treatment and completed it. There is no mention of any data of patients who had dropped out of treatment during the specified period. It would be pertinent to know the proportion of anxiety patients who started the treatment but did not complete it.

6. PLOS authors have the option to publish the peer review history of their article (what does this mean?). If published, this will include your full peer review and any attached files.

Reviewer #1: No

Reviewer #2: No

---

## [Author Response · Author response to Decision Letter 0]

19 Sep 2023

Reviewer #1: 

Please adding more relevant key words

- Thank you, we have added several key words.

Please consistently use same format in describing the dates such as July 1st 2020 throughout the article.

- We are grateful for this point, and have made the dates consistent throughout the manuscript.

The result and discussion only highlighted about the efficacy of CBT for anxiety but in the methodology indicated that the respondents also received DBT.

- CBT is a large conglomerate of therapy techniques that includes DBT. However, we appreciate Reviewer #1’s point and have mentioned these approaches separately in the paper.

Please mention how many sessions your respondents received and is there any standardized protocol to for your intervention. Please discuss the medium that you used in conducting your intervention.

- We have added the number of weekly sessions completed by each study group to Table 1. We have also clarified in the introduction and methods that patients presented to a naturalistic outpatient setting. Finally, we have added to the discussion that this feature of our study adds to the ecological validity of the findings.

Please explain how you compare the unequal group size and controlled the possible bias.

- Multilevel modeling is robust to unequal group sizes. We have added a brief comment to this effect at the start of the Statistical Analyses section.

Reviewer #2: The role of CBT in anxiety treatment is an important area of research in mental health studies. The present article will add to the research in this area.

- Thank you for these kind comments.

There are a few clarifications needed from the authors:

1. Was there uniformity of treatment in all groups? Did the "during pandemic" and "post

pandemic " group receive any additional counselling, specific to the pandemic? Was their

counselling more frequent?

- Our results suggest that groups received a similar number of sessions, relative to their aggregate time in therapy.

2. The data in this study is of anxiety patients who adhered to treatment and completed it. There is no mention of any data of patients who had dropped out of treatment during the specified period. It would be pertinent to know the proportion of anxiety patients who started the treatment but did not complete it.

- As above comments to Reviewer #1, we have added the number of treatment sessions completed by each group. Note that the naturalistic nature of our study precludes assessment of “drop-outs” since there was no fixed start or end point for treatment.

---

## [Decision Letter · Decision Letter 1]

21 Dec 2023

Response to Anxiety Treatment Before, During, and After the COVID-19 Pandemic

PONE-D-23-19350R1

Dear Dr. Rosmarin,

We’re pleased to inform you that your manuscript has been judged scientifically suitable for publication and will be formally accepted for publication once it meets all outstanding technical requirements.

Kind regards,

Mohammad Farris Iman Leong Bin Abdullah, Dr Psych

Academic Editor

PLOS ONE

Additional Editor Comments (optional):

Reviewers' comments:

Reviewer's Responses to Questions

**Comments to the Author**

1. If the authors have adequately addressed your comments raised in a previous round of review and you feel that this manuscript is now acceptable for publication, you may indicate that here to bypass the “Comments to the Author” section, enter your conflict of interest statement in the “Confidential to Editor” section, and submit your "Accept" recommendation.

Reviewer #1: All comments have been addressed

Reviewer #2: All comments have been addressed

2. Is the manuscript technically sound, and do the data support the conclusions?

Reviewer #1: Yes

Reviewer #2: Yes

3. Has the statistical analysis been performed appropriately and rigorously? 

Reviewer #1: Yes

Reviewer #2: I Don't Know

4. Have the authors made all data underlying the findings in their manuscript fully available?

Reviewer #1: Yes

Reviewer #2: Yes

5. Is the manuscript presented in an intelligible fashion and written in standard English?

Reviewer #1: Yes

Reviewer #2: Yes

6. Review Comments to the Author

Reviewer #1: (No Response)

Reviewer #2: All queries in previous version have been satisfactorily addressed by the authors. No further queries from this reviewer.

7. PLOS authors have the option to publish the peer review history of their article (what does this mean?). If published, this will include your full peer review and any attached files.

Reviewer #1: No

Reviewer #2: No

---

## [Editor Report · Acceptance letter]

19 Feb 2024

PONE-D-23-19350R1 

PLOS ONE

Dear Dr. Rosmarin, 

I'm pleased to inform you that your manuscript has been deemed suitable for publication in PLOS ONE. Congratulations! Your manuscript is now being handed over to our production team.

Kind regards, 

on behalf of

Dr. Mohammad Farris Iman Leong Bin Abdullah 

Academic Editor

PLOS ONE